# Correlation between Cerebrospinal Fluid IL-12 Levels and Severity of Encephalopathy in Children

**DOI:** 10.3390/jcm10173873

**Published:** 2021-08-28

**Authors:** Hisashi Kawashima, Shinichiro Morichi, Gaku Yamanaka, Hiroo Terashi, Yasuyo Kashiwagi

**Affiliations:** 1Department of Pediatrics and Adolescent Medicine, Tokyo Medical University, Tokyo 160-0023, Japan; shinichiro-m@hotmail.co.jp (S.M.); gaku@tokyo-med.ac.jp (G.Y.); hoyohoyo@tokyo-med.ac.jp (Y.K.); 2Department of Neurology, Tokyo Medical University, Tokyo 160-0023, Japan; terashi@tokyo-med.ac.jp

**Keywords:** influenza, VEGF, IL-7, NOx, IL-12 p70

## Abstract

The roles of cytokines in the cerebrospinal fluid (CSF) of patients with acute encephalopathy remain unclear and controversial. In this study, the profiles of 26 cytokines and others were determined in 17 children with infection-associated neurological complications. Interleukin (IL)-12 levels were found to be high in a few of the patients. A comparison of the IL-12 levels in the CSF of patients demonstrated that IL-12 (p70) is almost always increased in those with encephalopathy. Levels of IL-12 in the CSF were highly correlated with the levels of PDGF-bb and IL-RA. IL-12 levels were found to be weakly correlated with IFN-γ levels, and strongly correlated with VEGF levels. These results demonstrate that IL-12 levels may affect the clinical symptoms of pediatric patients with encephalopathy.

## 1. Introduction

Encephalopathy is an acute disease that is often accompanied with a viral infection. The influenza virus and HHV-6 are major causes of infant viral infections, and are also known to be associated with central nervous system (CNS) symptoms, such as seizures [1]. Several studies have reported that viruses can be detected in the cerebrospinal fluid (CSF) of some patients who have encephalopathy [2]. However, in most cases, a causative agent cannot be detected in the CSF or brain tissue, even by highly sensitive methods. Recent studies have shown that cytokines play an important role in acute encephalopathy [3,4,5]. Interleukin (IL)-12 has been reported to increase the morbidity associated with viral encephalitis, by increasing the ratio of the damaging cytokine IFN-γ to the protective cytokine IL-10 [6]. IL-12 and IL-23 are bioactive cytokines, and their biological functions are becoming clear. In experimental autoimmune encephalomyelitis, which is an animal model of the autoimmune disease multiple sclerosis, anti-IL12 antibodies that block IL-12 were found to prevent disease relapse [7]. In this study, we analyzed the levels of IL-12 in the CSF of pediatric patients with encephalopathy, to clarify the clinical significance of IL-12 in pediatric encephalopathy.

## 2. Materials and Methods

In this study, acute encephalopathy was used as a generic term for acute brain dysfunction, usually preceded by infection. Its main symptoms are impaired consciousness and signs of increased intracranial pressure, often accompanied with convulsions or seizures, as reported by Mizuguchi et al. [8]. Seventeen patients were enrolled in this study between January 2005 and January 2019 with the following infections: 11 with influenza, 2 with HHV-6, 2 with respiratory syncytial virus, 1 with rotavirus, and 1 unknown, who had seizures, consciousness disorder, and imaging findings that suggested acute encephalopathy. The subjects consisted of 11 boys and 6 girls, aged from 10 months to 8 years. All patients experienced unconsciousness and/or generalized seizures, followed by epileptic focal seizures (4 patients had generalized seizures). CSF was obtained from the patients on the day of admission or the day after. Leukocytosis in the CSF was not observed in any of the patients. CSF samples were all negative for the usual bacterial cultures. The patients’ encephalopathy types were the following: acute necrotizing encephalopathy (ANE) in 1 patient, mild encephalitis/encephalopathy with a reversible splenial lesion (MERS) in 1 patient, exotoxic types in 3 patients, and untypable in 1 patient [8]. Five age-matched patients with purulent meningitis, as well as 15 patients with seizures that were not a result of infection (11 with West syndrome, 1 with Dravet syndrome, and 3 with other types of epilepsy) were enrolled as controls.

The concentrations of the following 27 cytokines were measured using the Bio-Plex Multiplex Cytokine Assay: interleukin (IL)-1β, IL-RA, IL-2, IL-4, IL-5, IL-6, IL-7, IL-8, IL-9, IL-10, IL-12 (p70), IL-13, IL-15, IL-17, exotoxin, granulocyte colony-stimulating factor (G-CSF), granulocyte macrophage colony-stimulating factor (GM-CSF), interferon (INF)-γ, IP-10, monocyte chemoattractant protein (MCP)-1, macrophage inflammatory protein (MIP)-1β, RANTES, PDGF-bb, VEGF, FGF, and tumor necrosis factor (TNF)-α. Statistical analyses were performed using IBM^®^ SPSS^®^ Statistics version 25.0 software (New York, United States). A Pearson *r*-coefficient of greater than 0.8 was considered to indicate a statistically significant correlation (*p*-value < 0.05).

## 3. Results

There was no correlation between a patient’s serum level and CSF level of IL-12, as shown in Figure 1. IL-12 (p70), which is a heterodimer of p40 and p35, is almost always increased in the CSF of patients with encephalopathy. The levels of IL-12 (p70) in patients with encephalopathy were statistically higher than those in patients with West syndrome and convulsive diseases (Figure 2). IL-12 (p70) levels in the CSF strongly correlated with VEGF levels, as shown in Figure 3. The levels of IL-12 (p70) in the CSF were highly correlated with PDGF-bb (r = 0.79) and IL-RA (r = 0.825). IL-12 (p70) levels were weakly correlated with IFN-γ (r = 0.562) levels (Table 1). We compared the outcomes of the patients with influenza encephalopathy for whom we were able to obtain cytokine profiles. IL-12 (p70) levels did not correlate with the prognosis of the patients (Figure 4).

The line represents approximate straight line (R is coefficient of determination).

Good indicates no sequelae, and poor indicates death or being bedridden. The boxes in the graph indicate the first quartile and third quartile, and the bars indicate the maximum, median, and minimum values. Statistical analysis was performed using the unpaired *t*-test. There was no statistically significant difference between the two groups.

## 4. Discussion

IL-12 was initially identified as a factor produced by human Epstein–Barr virus-transformed B-cell lines. Cytotoxic lymphocyte maturation factor is a cytokine that synergized with IL-2 in the induction of lymphokine-activated killer cells and cytotoxic T lymphocytes. IL-12 promotes the production of IFN-γ and TNF-α from T-cells and NK cells, and activates NK cells and cytotoxic T-lymphocytes, which enhance cytotoxicity. Subsequently, IL-12 was found to have two subunits, i.e., p35 and p40. p35 is constantly produced, and p40 is produced upon the activation of macrophages, dendritic cells, and B-cells, but not T-cells. IL-12 usually functions as a heterodimer of p40 and p35 (p70), but can also form a homodimer of p40, which inhibits p70 activity. IL-12 p40 production requires the interaction of CD40/CD40L with T-cells and IFN-γ. This production is promoted by GM-CSF, and is suppressed by TGF-β, IL-4, IL-10, and IL-13. The IL-12 receptor (IL-12R) consists of β1 and β2 chains, and Tyk2 and Jak2, which activate STAT4, are associated with the intracellular region of the β2 chain. IL-12 mainly differentiates Th0 cells into Th1 cells. The expression of the IL-12R is essential for Th1 cell differentiation, which is induced by IL-2 [9]. In the present study, we measured the levels of IL-12 p70, which reflects the activity of p40. A recent study demonstrated that patients with multiple sclerosis have high levels of IL-12 p40. Therefore, IL-12 is expected to be a useful biomarker for determining the appropriate treatment for patients with multiple sclerosis [10].

There have been many reports regarding the role of IL-12 in infectious CNS diseases. Stahel et al. reported that the mean IL-12 CSF level and IL-12 CSF/serum ratio were significantly increased in 10 patients in the course of 14 days after severe head trauma, compared with CSF samples from 15 control patients [11]. Komatsu et al. analyzed the role of IFN-γ in a host’s defense mechanism against exogenous IL-12 as well as the clearance of vesicular stomatitis virus (VSV) from the CNS, and found that IL-12 treatment resulted in substantially decreased VSV titers in brain homogenates, as well as decreased immunohistochemical detection of VSV antigens in tissue sections. In vitro studies in which purified IL-12 or IFN-γ were added to the cell culture medium demonstrated the induction of nitric oxide synthase isoforms in neurons, glia, and macrophages, and MHC II on glia and macrophages [12]. IL-12 has a substantial effect on promoting the survival and recovery of mice infected with VSV, when administered at the time of infection, as well as after infection [13]. Primed T-cells directly act to clear measles virus infections from the brain by their immune effector function via a noncytolytic IL-12- and IFN-γ-dependent mechanism [14]. Vonder et al. analyzed cytokine levels within the frontal cortex of rats with severe brain injury, and demonstrated a selective increase in IL-12, which was associated with the magnitude of change in impulsive choices caused by both mild and severe traumatic brain injury. They concluded that in addition to brain tissue loss, inflammatory pathways mediated by IL-12 contributed to the increased impulsivity [15].

In this study, children with encephalopathy showed an increase in IL-12 levels. The patients comprised children with ANE, MERS, and exotoxic types of encephalopathy, who displayed high signals on MR imaging in the acute phase, and intractable seizures. These clinical symptoms are consistent with the reported biological phenomenon of IL-12. We previously reported that NOx is overproduced in children with encephalopathy [16,17]. Bombeiro et al. also showed that the overproduction of NO may account for the neuronal death that occurs in *T. cruzi*-infected IL-12 p40 KO mice [18]. Therefore, IL-12 may play a crucial role in the pathophysiology of encephalopathy in children through NO production.

Alternatively, in the present study, IL12 (p70) levels were found to have a strong correlation with VEGF levels, which affect the permeability of the blood–brain barrier, (BBB). Hossain reported that cerebellar VEGF expression was increased approximately twofold concurrent with the development of cerebellar microvascular hemorrhage, enhanced vascular permeability to serum albumin, and vasogenic cerebellar edema in acute lead intoxication [19]. Patients with influenza A/H1N1 infection and acute respiratory distress syndrome/acute kidney injury have an overproduction of MCP-1, VEGF, and IP-10, possibly contributing to kidney injury, and are reported to be associated with a higher risk of death [20].

The enhanced viral clearance and promotion of host recovery by IL-12 treatment has been described along with its implications in the treatment of human encephalitis [21]. A study on the effects of IL-12/23 p40 neutralizing antibodies in patients with relapsing-remitting advanced multiple sclerosis demonstrated no clinical or radiological improvement in any treatment group compared with the placebo control group, although blocking these cytokines via a neutralizing antibody caused substantial improvements in animal models of the disease. The authors hence concluded that the antibodies in a more limited subset of patients with very early disease might show a treatment effect [22]. In our study, IL12 (p70) did not correlate with the prognosis of patients with influenza encephalopathy. A host’s response to infection is also regulated by the sex of the host, and the age at infection. Specific mucosal humoral immunity and systemic cellular immunity are also thought to play roles in the prevention of infection.

Although IL-1 beta and its antagonists have been reported to be associated with West syndrome [23], there are very few studies on the association between IL-12 and epilepsy. In our study, we did not detect increased levels of IL-12 in the CSF of patients with West syndrome or other types of epilepsy. Therefore, autoimmune factors might not play a substantial role in children with West syndrome and other types of epilepsy.

To develop a better understanding of cytokines, the pathophysiology of acute encephalopathy should be elucidated. Therapeutic efficacy is very poor for patients in the late stages of encephalopathy [23]. However, if patients in the early stages of disease are provided with appropriate treatments, there is a possibility that they will be cured. Anti-IL-12 treatments and other therapies available for patients in the early stages of disease are hence expected to be effective for curing encephalopathy, as well as for alleviating the associated sequalae.

A limitation of this study is that the sample size of patients with encephalopathy caused by virus was too small to demonstrate statistically significant data to support our conclusion, particularly regarding the effects on outcome.

## Figures and Tables

**Figure 1 jcm-10-03873-f001:**
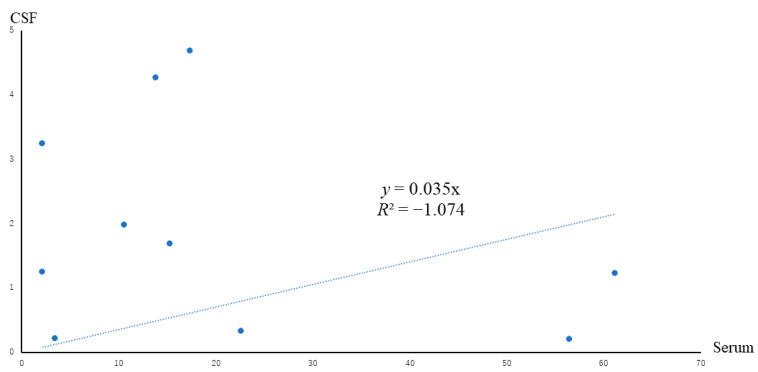
Blue dots indicate IL-12 (p70) levels in the serum and CSF obtained from the same patients.

**Figure 2 jcm-10-03873-f002:**
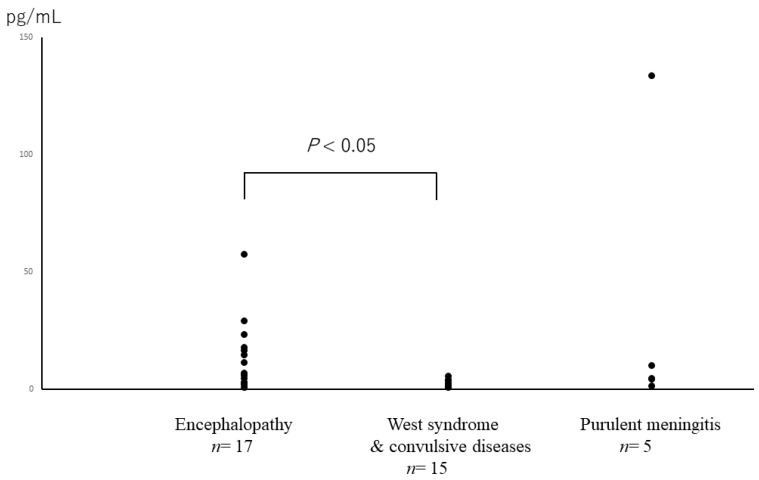
IL-12 (p70) levels in the CSF obtained from patients with different diseases.

**Figure 3 jcm-10-03873-f003:**
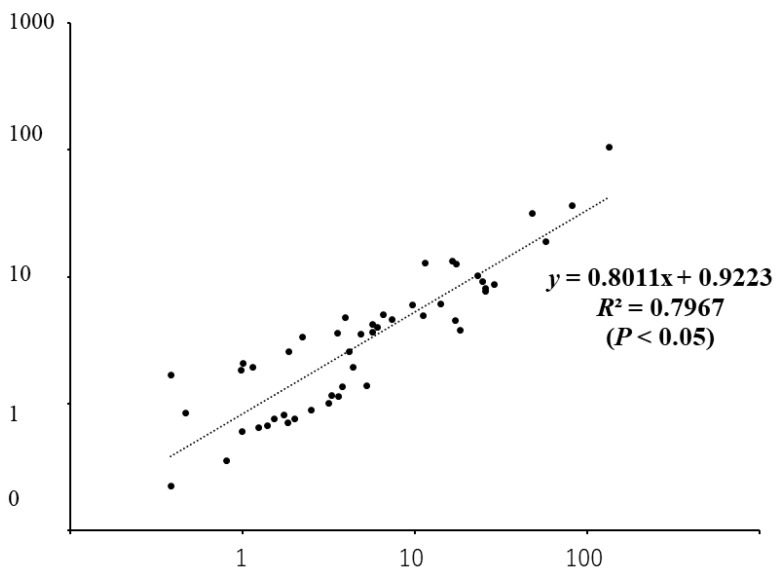
IL12 (p70) and VEGF levels in cerebrospinal fluid obtained from all patients.

**Figure 4 jcm-10-03873-f004:**
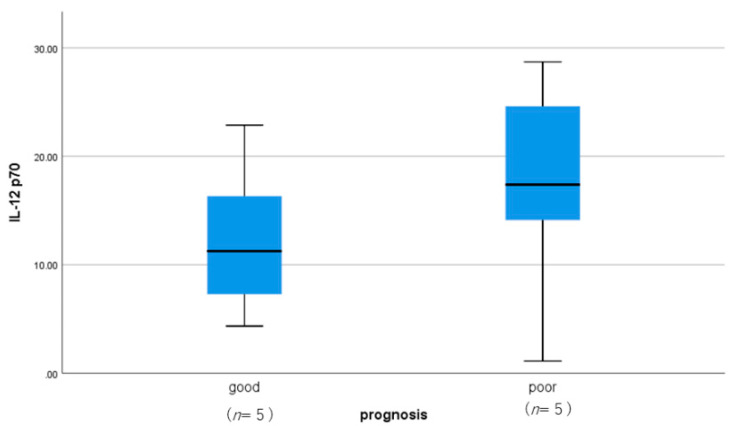
Comparison of CSF IL12 (p70) levels in patients with influenza-associated encephalopathy with good and poor prognoses.

**Table 1 jcm-10-03873-t001:** Correlation between CSF levels of IL-12 (p70) and various cytokines in patients with encephalopathy.

		IL-1b	IL-1ra	IL-2	IL-4	IL-5		IL-6	IL-7	IL-8		IL-9	IL-10	IL-13	IL-15
IL-12 (p70)	Pearson *R*	**0.643 ****	**0.825 ****	0.294 *	0.432 **	0.714 **		0.436 **	0.683 **	0.380 **		0.485 **	**0.781 ****	0.065	−0.095
	*p*-value	0.000	0.000	0.041	0.002	0.000		0.002	0.000	0.007		0.000	0.000	0.655	0.516
		IL-17	IFN-ϒ	IP-10	Exotaxin	FGF basic	PDGF-bb	VEGF	G-CSF	GM-CSF	MCP-1	MIP-1a	MIP-1b	RANTES	TNF-α
IL-12 (p70)	Pearson *r*	−0.124	0.562 **	0.203	**0.756 ****	0.037	**0.791 ****	**0.893 ****	0.415 **	−0.690 **	−0.068	0.257	0.779 **	0.674 **	0.669 **
	*p*-value	0.396	0.000	0.162	0.000	0.799	0.000	0.000	0.003	0.000	0.641	0.075	0.000	0.000	0.000

******p* < 0.01, ******
*p* < 0.05; *r* and *p* represent the Pearson *r*-correlation.

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
