# Peer review of "Correlation between Cerebrospinal Fluid IL-12 Levels and Severity of Encephalopathy in Children"

_jcm, 2021, doi:10.3390/jcm10173873_

Round 1

Reviewer 1 Report

The study made by Authors proposes new interesting physiopathological and therapeutical insights in the field of encephalopathy, a still broad-based concepts, potentially related with the most different ethiopatogenetical causes.

I think the paper deserves the publication, with some minor changes, that I propose:

  • I would add in methods section (abstract), the sample size); "infection associated neurological complications (row 18) can perhaps seem a bit confusing, so more clarity may benefit the paper readability
  • row 23: influenza associated encephalopathy? other viruses? i think in abstract, if only related to influenza, this can be omitted
  • row 31: "systemic" seems a little bit confusing
  • i would change convulsion/s into seizure/s
  • i would clarify encephalopathy, given the potential "direct" or "indirect" ethiopatogenesis (i would also add here a definition that may remain in the whole paper)
  • i would change partial seizures into focal seizures
  • raw 45: i think it should more explicitly stated that patients had to satisfy all the criteria proposed for encephalopathy (AND)
  • in methods section, when possibile, i think it could be interesting the timing of csf testing in the disease history
  • row 65: west syndrome, convulsive disease sound maybe a little confusing (epileptic syndromes? west syndrome and other epileptic syndrome are together considered?)
  • in results, when possibile, i would add some data with timing of samples, and a comparison with healthy aged patients data (when possible)
  • row 70-71 Figure 4: not clear if the correlation was analysed for influenza encephalopathy or other virus (see also figure 4)
  • figure 3: west syndrome? epileptic syndromes? 13 or 14 patients?
  • row 102: i would change EB into Epstein Barr for more clarity
  • discussion: minor changes, if possibile, to add more fluidity to the text, for example when switching to multiple sclerosis paragraph
  • in discussion, eventually add the data for csf sampling timing in our sample, when discussing therapeutic opportunities
  • row 141: i would change acute lead patients into acute lead intoxication (isn´t it?)
  •  

Author Response

Dear Reviewers,

We thank you for your comments and suggestions, which have helped us to improve our manuscript. We also appreciate the time you have given us to revise the manuscript. Our point-by-point responses to the comments are shown below. We hope that our revised manuscript is now suitable for publication in Journal of Clinical Medicine.

Responses to the comments by Reviewer 1

 Comments and Suggestions for Authors

 The study made by Authors proposes new interesting physiopathological and therapeutical insights in the field of encephalopathy, a still broad-based concepts, potentially related with the most different etiopathogenetic causes.

I think the paper deserves the publication, with some minor changes, that I propose:

  1. I would add in methods section (abstract), the sample size); "infection associated neurological complications (row 18) can perhaps seem a bit confusing, so more clarity may benefit the paper readability

Response:

To make this point clear, we added the number of subjects to this sentence, as follows.

 “In this study, the profiles of 26 cytokines and others were determined in 17 children with infection-associated neurological complications.” (page 1, line 17)

 row 23: influenza associated encephalopathy? other viruses? I think in abstract, if only related to influenza, this can be omitted

Response:

In this study, we compared the correlation between IL-12 level and the prognosis of children with influenza-associated encephalopathy. Therefore, we deleted this sentence from the revised manuscript.

  1. row 31: "systemic" seems a little bit confusing

Response:

To avoid confusion, we deleted this word from this sentence. (page 1, line 31)

  1. I would change convulsion/s into seizure/s

Response:

We changed the word “convulsion/s” to “seizure/s” throughout the manuscript (page 1, line 33; page 2, line 45; page 1, lines 33, 47, 48, and 52; and page 3, line 132).

  1. I would clarify encephalopathy, given the potential "direct" or "indirect" ethiopatogenesis (I would also add here a definition that may remain in the whole paper)

Response:

The definition of encephalopathy that we used excludes patients with inflammation of the central nervous system, such as leukocytosis in the cerebrospinal fluid. Therefore, the subjects in this study all had encephalopathy caused by an indirect etiopathogenesis. To make this point clear, we added the definition of encephalopathy to the revised manuscript, as follows.

“In this study, acute encephalopathy was used as a generic term for acute brain dysfunction, usually preceded by infection. Its main symptoms are impaired consciousness and signs of increased intracranial pressure, often accompanied with convulsions or seizures, as reported by Mizuguchi et al. [8]. ” (page 2, line 44)

  1. I would change partial seizures into focal seizures

Response:

In accordance with the comment, we changed the phrase “partial seizures” to “focal seizures”. (page 2, line 50)

  1. raw 45: I think it should more explicitly stated that patients had to satisfy all the criteria proposed for encephalopathy (AND)

Response:

Please see our response to comment no. 5.

  1. in methods section, when possible, I think it could be interesting the timing of CSF testing in the disease history

Response:

We added a sentence describing the timing of CSF testing to the revised manuscript, as follows.

“The CSF were obtained on first or second days after admission.” (page 2, line 51)

  1. row 65: west syndrome, convulsive disease sound maybe a little confusing (epileptic syndromes? west syndrome and other epileptic syndrome are together considered?)

Response:

Acute encephalopathy is usually preceded by infection. Its main symptoms are impaired consciousness and signs of increased intracranial pressure, often accompanied with convulsions or seizures. Therefore, we compared the CSF of children with frequent seizures and those with intractable seizures as controls.

  1. in results, when possible, I would add some data with timing of samples, and a comparison with healthy aged patients data (when possible)

Response:

We added the timing at which we obtained the CSF samples to the revised manuscript.

“CSF was obtained from the patients on the day of admission or the day after.”(page2 line 51)

Unfortunately, CSF samples from healthy children were not available, so we were unable to make this comparison.

  1. row 70-71 Figure 4: not clear if the correlation was analyzed for influenza encephalopathy or other virus (see also figure 4)

Response:

We compared the outcomes only of patients with influenza-associated encephalopathy. To make this point clear, we changed the sentence as follows in the Results section of the revised manuscript.

“We compared the outcomes of patients with influenza encephalopathy from whom we were able to obtain their profiles. IL-12 (p70) levels did not correlate with the prognosis of patients (Figure 4).” (page 2, line 72)

  1. figure 3: west syndrome? epileptic syndromes? 13 or 14 patients?

Response:

In Figure 3, we showed the data of patients with acute encephalopathy, West syndrome, intractable convulsive diseases, and purulent meningitis. Only 1 patient with West syndrome showed an exceptionally high level of IL-12. We omitted this patient from the data, and remade Figure 3.

  1. row 102: i would change EB into Epstein Barr for more clarity

Response:

In accordance with the comment, we changed “EB” to “Epstein Barr” in the revised manuscript.

  1. discussion: minor changes, if possible, to add more fluidity to the text, for example when switching to multiple sclerosis paragraph

Response:

In the Discussion section, we added a description regarding the treatment of multiple sclerosis using anti-IL-12 neutralizing antibodies, as follows.

“The enhanced viral clearance and promotion of host recovery by IL-12 treatment has been described along with its implications in the treatment of human encephalitis [21]. According to the study about IL12/23 p40 neutralizing antibody in patients with relapsing-remitting advanced multiple sclerosis, no clinical or radiologic improvement in any treatment group compared with placebo controls was reported although the blocking these cytokines via a neutralizing antibody caused dramatic improvements in animal models of the disease. They concluded that the antibodies in a more limited subset of patients with very early disease might show a treatment effect [22].” (page 4, line 144)

  1. in discussion, eventually add the data for csf sampling timing in our sample, when discussing therapeutic opportunities

Response:

In accordance with the suggestion, we added the following sentence suggesting future therapies to the revised manuscript.

“Anti-IL-12 treatments and other supported therapies in early stage are hence expected to be effective for curing encephalopathy, as well as to alleviate the associated sequalae.” (page 4, line 162)

  1. row 141: i would change acute lead patients into acute lead intoxication (isn´t it?)

Response:

We apologize for our careless mistake. We corrected this in the revised manuscript.

Reviewer 2 Report

In the present paper, authors deal with a potential correlation between cerebrospinal fluid IL-12 levels and severity of  encephalopathy in children.  In this aim, they looked for CSF levels of  27 citokines and other immune- mediate factors. They have compared 18 pts with various acute infectious diseases including influence, rotavirus and HHV-6.2 viruses with a control group mainly affected by epileptic encephalopathies like West syndrome, Dravet syndrome and others.

The study is interesting, though sample size is honestly very small to try to demonstrate any significant correlation. There are, furthermore, some criticisms authors should try to deal with:

  • the term “encephalopathy” is used to mean acute encephalopathies? Why if an encephalopathy may be chronic or subacute or progressive?
  • The increase of IL 12 is associated with viral infectious diseases versus non infectious: this is not very new.
  • Considering the potential role of autoimmune factors in epileptogenesis, authors should comment on the results obtained in the control group in epileptic patients, in the light of literature data.
  • Possibly, the timing of CSF sample in each patient should be specified; might it influence ILs levels?
  • Authors should finally better clarify what this study adds to the existing literature.

Author Response

Reviewer 2

Comments and Suggestions for Authors

In the present paper, authors deal with a potential correlation between cerebrospinal fluid IL-12 levels and severity of encephalopathy in children. In this aim, they looked for CSF levels of 27 cytokines and other immune-mediate factors. They have compared 18 pts with various acute infectious diseases including influence, rotavirus and HHV-6.2 viruses with a control group mainly affected by epileptic encephalopathies like West syndrome, Dravet syndrome and others.

The study is interesting, though sample size is honestly very small to try to demonstrate any significant correlation. There are, furthermore, some criticisms authors should try to deal with:

  1. the term “encephalopathy” is used to mean acute encephalopathies? Why if an encephalopathy may be chronic or subacute or progressive?

 Response:

As we focused on the role of cytokines and others in the acute disease state in this study, patients with acute infection-associated encephalopathy and intractable seizures were included as subjects. To make this point clear, we added the definition of acute encephalopathy to the revised manuscript, as follows.

“In this study, acute encephalopathy was used as a generic term for acute brain dysfunction, usually preceded by infection. Its main symptoms are impaired consciousness and signs of increased intracranial pressure, often accompanied with convulsions or seizures, as reported by Mizuguchi et al. [8]. ” (line 44)

  1. The increase of IL 12 is associated with viral infectious diseases versus non infectious: this is not very new.

Response:

In this study, we focused on IL-12 and its correlation with other factors (disease outcome, other cytokines, and convulsive diseases), which has not been analyzed in detail previously. Moreover, we found no increase in CSF IL-12 levels of patients with purulent meningitis, which is a new finding.

  1. Considering the potential role of autoimmune factors in epileptogenesis, authors should comment on the results obtained in the control group in epileptic patients, in the light of literature data. 

Response:

Many studies have reported an association between IL-1 and West syndrome. However, there are very few studies on the association between epilepsy and IL-12. In our present study, we did not focus on the potential role of autoimmune factors in the epileptogenesis of patients with West syndrome and other types of epilepsy.To make this point clear, we added the following sentences to the Discussion section of the revised manuscript.

“Although IL 1beta and antagonist have been reported to correlate in West syndrome [23], there are very few in epileptic patients about IL-12. In our study IL 12 in CSF in West syndrome and other epileptic patients did not show increasing. The potential role of autoimmune factors in epileptogenesis might be not the potential role in West syndrome and other epileptic children.” (line 155)

  1. Possibly, the timing of CSF sample in each patient should be specified; might it influence ILs levels?

Response:

To make this point clear, we added the following sentence to the revised manuscript.

“The CSF were obtained on first or second days after admission.” (page 2, line 51)

  1. Authors should finally better clarify what this study adds to the existing literature.

Response:

To clarify what our study adds to the existing literature, we included a discussion about future treatments, as well as a new reference, as follows.

“The enhanced viral clearance and promotion of host recovery by IL-12 treatment has been described along with its implications in the treatment of human encephalitis [21]. According to the study about IL12/23 p40 neutralizing antibody in patients with relapsing-remitting advanced multiple sclerosis, no clinical or radiologic improvement in any treatment group compared with placebo controls was reported although the blocking these cytokines via a neutralizing antibody caused dramatic improvements in animal models of the disease. They concluded that the antibodies in a more limited subset of patients with very early disease might show a treatment effect [22]. In our study IL12 (p70) did not correlate with the prognosis of patients with influenza encephalopathy. A host’s response to infection is also regulated by the sex of the host, and the age at infection. Specific mucosal humoral immunity and systemic cellular immunity are also thought to play roles in the prevention of infection.” (page 3, line 146)

“However, if patients in the early stages of disease are provided with appropriate treatments, there is a possibility that they will be cured. Anti-IL-12 treatments and other supported therapies in early stage are hence expected to be effective for curing encephalopathy, as well as to alleviate the associated sequalae.” (page 4, line 158)